# Giant barocaloric effect in the ferroic organic-inorganic hybrid [TPrA][Mn(dca)$_3$] perovskite under easily accessible pressures

Juan M. Bermúdez-García[1], Manuel Sánchez-Andújar[1], Socorro Castro-García[1], Jorge López-Beceiro[2], Ramón Artiaga[2] & María A. Señarís-Rodríguez[1]

The fast growing family of organic–inorganic hybrid compounds has recently been attracting increased attention owing to the remarkable functional properties (magnetic, multiferroic, optoelectronic, photovoltaic) displayed by some of its members. Here we show that these compounds can also have great potential in the until now unexplored field of solid-state cooling by presenting giant barocaloric effects near room temperature already under easily accessible pressures in the hybrid perovskite [TPrA][Mn(dca)$_3$] (TPrA: tetra-propylammonium, dca: dicyanamide). Moreover, we propose that this will not be an isolated example for such an extraordinary behaviour as many other organic–inorganic hybrids (metal-organic frameworks and coordination polymers) exhibit the basic ingredients to display large caloric effects which can be very sensitive to pressure and other external stimuli. These findings open up new horizons and great opportunities for both organic–inorganic hybrids and for solid-state cooling technologies.

[1] QuiMolMat Group, Department of Chemistry, Faculty of Science and Advanced Scientific Research Center (CICA), Zapateira, University of A Coruna, 15071 A Coruna, Spain. [2] Department of Naval and Industrial Engineering, Esteiro, University of A Coruna, 15471 Ferrol, Spain. Correspondence and requests for materials should be addressed to J.M.B.-G. (email: j.bermudez@udc.es) or to M.A.S.-R. (email: m.senaris.rodriguez@udc.es).

Refrigeration and air conditioning already represents more than 20% of the world's energy consumption and demand in these fields is expected to grow dramatically in the coming decades. Nevertheless, the conventional cooling technology that is used nowadays suffers from a relatively low efficiency and relies on the vapour compression of gases that have been proven to be hazardous and/or pollutants, such as the greenhouse gases (hydrochlorofluorocarbons and hydrofluorocarbons), which in turn are going to be prohibited in Europe in 2020 (EU Regulation No 517/2014).

Therefore, finding environmentally friendly and more efficient cooling alternatives constitutes a major issue and concern, and one of the present big scientific and technological challenges.

In that context, a promising approach is based on solid-state materials exhibiting large caloric effects[1–3], and in which the refrigeration capacity is associated with a large isothermal entropy change or with a large adiabatic temperature change induced by different external stimuli[1–6], such as mechanical stress—namely, uniaxial strain (the so-called elastocaloric effect)[7] or hydrostatic pressure (barocaloric effect)[8]—electric field (electrocaloric effect)[9,10], or magnetic field (magnetocaloric effect)[11–13], effects that are enhanced near to phase transitions[4].

Up to now, the most promising caloric materials are those exhibiting giant magnetocaloric effects[1,2]. However, they require the application of rather large magnetic fields ($H > 2$ T) and are rather expensive magnetic materials, many of them containing rare-earths, drawbacks for the wide industrial and commercial application of the resulting so-called magnetic refrigeration.

Another important obstacle that has been very recently detected[14] is the hysteretic effects many of them present under field cycling, and which can compromise their caloric performance.

On the other hand, materials that display solid-state caloric effects driven by applied pressure/stress could lead to more accessible and economic technological solutions. Nevertheless, less efforts have been devoted to their study because such effects were expected to be minimal in solids due to the small entropy changes and/or reduced pressure sensibility of substances in such aggregation state of the matter[4]. For example, in the case of the ceramic (inorganic) perovskite $Pr_{1-x}La_xNiO_3$ the application of hydrostatic pressures up to 5 kbar (0.5 GPa) is known to induce a small effective cooling, in competition with the elastic heating[15,16]; and in the related $PbTiO_3$ and $BaTiO_3$ perovskites those values have been predicted to be relatively small[17,18]. Nevertheless, quite recently, giant barocaloric effects have been found near room temperature associated with the giant magnetocaloric response of a few magnetic alloys[19,20], in some perovskite-like fluorides and oxyfluorides[21], and in ferrielectric ammonium sulphate[8], even if under the application of pressures $P > 0.1$ GPa.

What we report in this work is the finding of giant barocaloric effects near room temperature which can be already induced by considerable small pressures ($P < 0.007$ GPa) in a solid of formula $[TPrA][Mn(dca)_3]$ (TPrA = tetrapropylammonium cation, $(CH_3CH_2CH_2)_4N^+$; dca = dicyanamide anion, $[N(CN)_2]^-$) with perovskite-like structure. This compound belongs to the rather young, but very fast growing group of compounds widely known as organic–inorganic hybrids, which include the large family of metal-organic frameworks[22], and coordination polymers[23].

These hybrids, where the multiple possible combinations of inorganic and organic moieties allow for an enormous structural and chemical diversity[24], are arising great interest in many diverse fields such as gas adsorption or catalysis (in the case of hybrids with open porous structures); and more recently, in connection with remarkable functional and multifunctional properties displayed by members with denser structures, many of them belonging to the group of the so-called hybrid perovskites with general formula $ABX_3$. Remarkable examples are, for instance, $(CH_3NH_3)PbI_3$ and

related compounds[25,26], which exhibit exceptional optoelectronic properties for photovoltaic applications. And also the [AmineH][M(X)_3] perovskite families (AmineH = midsized protonated amines, M = divalent transition metal cations, X = bidentate-bridge ligands $HCOO^-$, $N_3^-$, $CN^-$, and so on) which display outstanding functionalities such as cooperative magnetic, electric and/or elastic order, and magnetically induced multiferroicity; moreover, they can also act as precursors for different nanostructured functional materials[27–38].

Nevertheless, an aspect that had not yet been explored in these hybrid materials is their potentiality as solid-state caloric materials, in view of the stimuli-driven caloric effects they can display.

For this purpose, we focus in the hybrid perovskite $[TPrA][Mn(dca)_3]$, where our recent studies have revealed a first-order phase transition at $T_t \sim 330$ K, which involves a complex structural transition that shows a large response towards pressure and temperature[36]. According to our previous characterization, both above and below the transition the compound shows a perovskite-like structure where the $Mn^{2+}$ cations, in an octahedral $[MnN_6]$ coordination, are bridged by dca ligands forming a 3D framework, while the TPrA cations are located in the resulting pseudo-cuboctahedral cavities. Nevertheless, polymorph I ($T < T_t$) shows a much more ordered crystal structure with less disorder in the dca ligands and the TPrA cations, which in addition show a cooperative antiferrodistortive displacement from the centre of the cavities. Meanwhile, polymorph II ($T > T_t$) shows a much larger thermal disorder in the C atoms of the TPrA-pending propyl groups and in the N and C atoms of the dca ligands and a centred distribution of the A-cations.

In order to study the barocaloric effect of this organic–inorganic hybrid compound, we use both quasi-direct methods, by carrying out isobaric calorimetric analysis at different low ($P < 0.007$ GPa) and higher pressures (up to 0.1 GPa); and direct methods, by carrying out cyclic isothermal calorimetric measurements, comparing such results with those obtained by indirect Clausius–Clapeyron estimations. And we demonstrate that this material displays a very large caloric effect of $37.0$ J kg$^{-1}$ K$^{-1}$, slightly above room temperature and under the application of very small applied pressures ($P < 0.007$ GPa), overpassing most of the best caloric materials[4]. Moreover, we address the issues of thermal hysteresis and reversible entropy changes in this material (30.5 J kg$^{-1}$ K$^{-1}$ at $\sim 0.007$ GPa) and propose the conditions for an efficient and reversible barocaloric response.

## Results

**Differential scanning calorimetry under applied pressure.** To study the barocaloric effect induced by external hydrostatic pressure in this compound, we have performed differential scanning calorimetry studies as a function of pressure (Pressure differential scanning calorimetry (P-DSC)), from 1 bar to 1,000 bar (that is from 0.0001 to 0.1 GPa), and as a function of temperature, from room temperature to 370 K (see Methods). The obtained results fully corroborate, in first place, the first-order nature of the phase transition, which displays a small hysteresis of $\sim 0.9$ K at a heating/cooling rate of 2 K min$^{-1}$. They also confirm the large entropy change of $\Delta S_{trans} \sim 42.5$ J kg$^{-1}$ K$^{-1}$ (see Supplementary Fig. 1) associated to this solid–solid phase transition and whose microscopic physical origin can, in principle, be made up of configurational, rotational and vibrational contributions[39].

Taking into account that if the crystal structures of the ordered and disordered phases are known, the configurational entropy can be calculated as $\Delta S_{config} = R \ln(N)$, with $N = (n_2/n_1)$, where $n_2$ and $n_1$ are the number of configurations in the two polymorphs

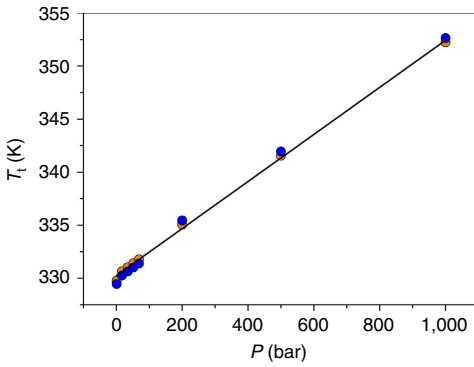

**Figure 1 | Barocaloric coefficient.** Pressure dependence of the transition temperature values ($T_t$) obtained from the onset temperatures of P-DSC analysis carried out under pressures from 1 to 1,000 bar both on heating (orange) and cooling (blue).

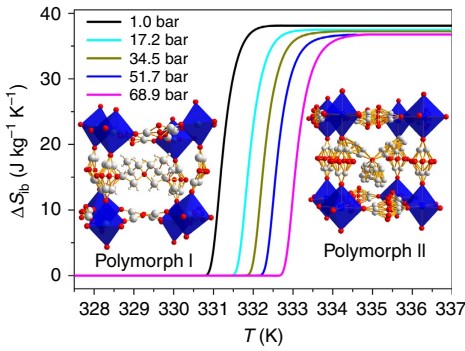

**Figure 2 | Isobaric entropy change.** Isobaric entropy change as a function of temperature in the low pressure range (1–68.9 bar), related to the first-order phase transition from polymorph I to II (as calculated from experiments carried out on heating).

and $R$ is the gas constant ($8.314\,\mathrm{J\,mol^{-1}\,K^{-1}}$) (ref. 39), for [TPrA][Mn(dca)$_3$] we have roughly estimated an $N \sim 10.3$ (see Supplementary Fig. 2) and a $\Delta S_{conf.} \sim 19.4\,\mathrm{J\,mol^{-1}\,K^{-1}}$ ($44.1\,\mathrm{J\,kg^{-1}\,K^{-1}}$). The fairly good agreement between this entropy value and the experimental one, $\Delta S_{trans.} \sim 42.5\,\mathrm{J\,kg^{-1}\,K^{-1}}$ (specially taking into account the simplicity of the model and the complexity of the structures) suggests that this configurational factor (the partial order/disorder of the dca and TPrA groups) is mainly determining the total entropy change in this compound.

The obtained results also indicate that the phase transition temperature gets progressively displaced towards higher temperatures when increasing pressure. From these data, we have calculated the so-called barocaloric coefficient ($\delta T_t/\delta P$) as the variation of the transition temperature as a function of pressure obtaining a value of $\delta T_t/\delta P = 23.1\,\mathrm{K\,kbar^{-1}}$ (see Fig. 1). This value is in good agreement with our previous results obtained by different techniques (P-DSC, dielectric measurements under pressure and pressure–volume–temperature—PVT—analysis)[36]. Very interestingly, the initial situation is recovered after decompression, even when performing several cycles with the material showing a reproducible pressure cyclability (see Supplementary Fig. 3). The reversibility of this process is also corroborated by powder X-ray diffraction (PXRD) data, which show that after decompression (from 68.9 bar and 1,000 bar, respectively) the material is still fully crystalline and polymorph I is the phase present at ambient conditions (see Supplementary Fig. 4).

**Isobaric entropy change at different pressures.** From the calorimetric data obtained in the experiments carried out under low ($P < 70$ bar) and higher pressures (up to 1,000 bar) (see Methods), it is possible to calculate the barocaloric effect of this compound. For this purpose, we have calculated the isobaric entropy change, $\Delta S_{ib}$, as a function of temperature at each pressure, using equation (1) (refs 3,40), where $Q(P)$ is the experimental heat flux measured at different pressures, $T'$ is the heating rate and $T$ is the temperature:

$$\Delta S_{ib} = S(T,P) - S(T_0, P) = \int_{T_0}^{T} \frac{1}{T}\frac{Q(P)}{T'}\,\mathrm{d}T \qquad (1)$$

In the low pressure region, the obtained data reveal that the entropy increases up to a plateau of $\sim 38.1\,\mathrm{J\,kg^{-1}\,K^{-1}}$ when the material undergoes the phase transition from polymorph I to polymorph II (see Fig. 2 and Supplementary Fig. 5a). Therefore,

this plateau sets the maximum value of entropy attainable in the caloric effect related to such phase transition.

In addition, and as it can be seen in Fig. 2, there is a systematic shift of the entropy curves towards higher temperatures as hydrostatic pressure increases. This is consistent with pressure enhancing the stability of the more ordered and lower volume polymorph I at the expense of the more disordered polymorph II.

The associated barocaloric effect for this material, which can be defined as the isothermal entropy change induced by hydrostatic pressure and calculated as the difference between the data at different pressures by this quasi-direct method, $\Delta S_{it(q-d)} = \Delta S_{ib}(P \neq 1) - \Delta S_{ib}(P = 1)$, is shown in Supplementary Figs 5b and 6a. As it can be seen there, the isothermal application of pressure up to 68.9 bar results in an entropy change of $\Delta S_{it(q-d)} = 37.0\,\mathrm{J\,kg^{-1}\,K^{-1}}$. Very interestingly, despite the relative small pressure applied ($P < 0.007$ GPa), the peak values are very close to the maximum obtainable caloric effect, $\Delta S_{ib} \sim 38.1\,\mathrm{J\,kg^{-1}\,K^{-1}}$, and appear slightly above room temperature.

Most remarkably, these large values compare well with those reported for giant magneto-, electro-, elasto-caloric materials, even if in those cases under the application of much larger fields (see Table 1).

We have also calculated the barocaloric effect of this material expressed in terms of the pressure-induced adiabatic temperature change, $\Delta T_{ad(q-d)}$, which is directly related to the isothermal entropy change, $\Delta S_{it(q-d)}$, through equation (2) (refs 21,41):

$$\Delta T_{ad(q-d)} = -(T/C_p)\Delta S_{it} \qquad (2)$$

where $C_p$ is the specific heat capacity, that we have experimentally measured ($C_p = 2,450\,\mathrm{J\,kg^{-1}\,K^{-1}}$, see Methods); and $T$ is the temperature at which the barocaloric effect occurs (here $T \sim 332$ K).

The obtained results, shown in Supplementary Fig. 6b, give a $\Delta T_{ad(q-d)} \sim 5.0$ K, value which is very large taking into account the relative low pressures used ($P = 68.9$ bar). This value is very competitive with those exhibited by other barocaloric materials under much higher pressures (see Table 1).

Moreover, we have performed studies at higher pressures, up to 1,000 bar. The calorimetric data obtained in this higher pressure region reveal a similar trend to that in the low pressure range. Nevertheless, it is important to note that the entropy starts to diminish at such pressure values (by $\sim 15\%$ at $P = 500$ bar, $\sim 32\%$ at $P = 1,000$ bar) (see Fig. 3 and Supplementary Fig. 7).

Such decrease is due to additional changes in isothermal entropy $\Delta S_+(P)$ associated with elastic heat, effects which are conventional, reversible and opposite in sign with respect to the

**Table 1 | Selected caloric parameters of some of the up-to-date best caloric materials.**

| Giant BC material | $T_t$ [K] | $|\Delta S_{it}|$ [J kg$^{-1}$ K$^{-1}$] | $|\Delta T_{ad}|$ [K] | $|\Delta P|$ [GPa] | RCP [J kg$^{-1}$] | RCP/$|\Delta P|$ [J kg$^{-1}$ GPa$^{-1}$] | Ref. |
|---|---|---|---|---|---|---|---|
| Ni$_{49.26}$Mn$_{36.08}$In$_{14.66}$ | 293 | 24 | 4.5 | 0.26 | 120 | 462 | 8,19 |
| LaFe$_{11.33}$Co$_{0.47}$Si$_{1.2}$ | 237 | 8.7 | 2.2 | 0.20 | 81 | 405 | 8,20 |
| Gd$_5$Si$_2$Ge$_2$ | 270 | 11 | 1.1 | 0.20 | 180 | 900 | 8,42 |
| Fe$_{49}$Rh$_{51}$ | 308 | 12.5 | 8.1 | 0.11 | 105 | 955 | 8,43 |
| Mn$_3$GaN | 285 | 21.6 | 4.8 | 0.09 | 125 | 1,389 | 8,44 |
| (NH$_4$)$_3$MoO$_3$F$_3$ | 297 | 55 | 15 | 0.5 | 5,200 | 10,400 | 21 |
| (NH$_4$)$_2$SO$_4$ | 219 | 60 | 8 | 0.1 | 276 | 2,760 | 8 |
| [TPrA][Mn(dca)$_3$] q-d methods* | 330 | 37.0 | 5.0 | 0.00689 | 66 | 9,518 | Herein |
| [TPrA][Mn(dca)$_3$] direct methods | 330 | 35.1 | 4.8 | 0.00689 | 62 | 9,089 | Herein |
| [TPrA][Mn(dca)$_3$] reversible† | 330 | 30.5 | 4.1 | 0.00689 | 54 | 7,896 | Herein |
| **Giant MC material** | **$T_t$ [K]** | **$|\Delta S|_{it}$ [J kg$^{-1}$ K$^{-1}$]** | **$|\Delta T|_{ad}$ [K]** | **$|\Delta H|$ [T]** | **RCP [J kg$^{-1}$]** | **RCP/$|\Delta H|$ [J kg$^{-1}$ T$^{-1}$]** | **Ref.** |
| Gd | 296 | 11 | 11 | 5 | 780 | 156 | 21 |
| Fe$_{1-x}$Rh$_x$ | 313 | 55 | 33 | 5 | 900 | 180 | 21 |
| Pr$_{0.63}$Sr$_{0.37}$MnO$_3$ | 300 | 8.52 | 5.7 | 5 | 511 | 102 | 45 |
| Gd$_5$Si$_2$Ge$_2$ | 276 | 18.4 | 15 | 5 | 535 | 107 | 45 |
| **Giant EC material** | **$T_t$ [K]** | **$|\Delta S|_{it}$ [J kg$^{-1}$ K$^{-1}$]** | **$|\Delta T|_{ad}$ [K]** | **$|\Delta E|$ [kV cm$^{-1}$]** | **RCP [J kg$^{-1}$]** | **RCP/$|\Delta E|$ [J cm kg$^{-1}$ kV$^{-1}$]** | **Ref.** |
| PbZr$_{0.95}$Ti$_{0.05}$O$_3$ | 500 | 8 | 12 | 480 | 1,080 | 2.25 | 9,21 |

BC, barocaloric; EC, electrocaloric; MC, magnetocaloric; RCP, relative cooling power; RCP/$|\Delta P|$, relative cooling power normalized per pressure unit; RCP/$|\Delta H|$, relative cooling power normalized per magnetic field unit; RCP/$|\Delta E|$, relative cooling power normalized per electric field unit; $T_t$, transition temperature; $|\Delta P|$, pressure change; $|\Delta H|$, magnetic field change; $|\Delta E|$, electric field change; $|\Delta S_{it}|$, isothermal entropy change; $|\Delta T_{ad}|$, adiabatic temperature change.
*For comparison purposes we have used the values obtained by quasi-direct methods of $\Delta S_{it(q-d)}$ and $\Delta T_{ad(q-d)}$ without taking into account the hysteresis losses.
†We have also included the value of $\Delta S_{it(rev)}$ and $\Delta T_{ad(rev)}$ that could be used for practical applications under the application of 68.9 bar.

pressure-driven isothermal entropy changes associated with the structural phase transition. This isothermal entropy $\Delta S_+(P)$ can be estimated away from the first-order transition via $\Delta S_+(P) = -[m^{-1}(\delta V/\delta T)_{P=0}]P$ (see Methods). Supplementary Fig. 8 shows the temperature dependence of $\Delta S_+(P)$ on applying pressure $P$. As it can be seen there, such contribution is almost negligible for $P < 68.9$ bar, with values which are around 30 times lower than those of the barocaloric effect related to the phase transition. Nevertheless for $P > 200$ bar, $\Delta S_+(P)$ becomes more significant, reaching at $P = 1,000$ bar and $T = 358$ K, value that is already $\sim 32\%$ of the entropy change associated with the structural phase transition.

As a result the barocaloric effect under higher pressures ($P > 200$ bar) decreases and turns to be $\Delta S_{it(q-d)} \sim 32.9$ J kg$^{-1}$ K$^{-1}$ ($P = 500$ bar) and $\Delta S_{it(q-d)} \sim 26.5$ J kg$^{-1}$ K$^{-1}$ ($P = 1,000$ bar) (see Fig. 3 and Supplementary Fig. 7).

On the other hand, and as indicated in the introduction, the thermal hysteresis associated with first-order phase transitions can reduce the stimuli-driven caloric effects under cyclic conditions[14], and become a main drawback for the efficiency of solid-state cooling devices. In this context and following the literature[14], we have recalculated for this organic–inorganic hybrid compound the reversible entropy changes that could be used for cooling, by taking into account the data obtained in both the heating and cooling cycles. Such reversible entropy changes correspond to the grey shaded area shown in Fig. 4 for different applied pressures.

As it can be seen there, for $P = 68.9$ bar, the attainable reversible entropy change gets reduced by a $\sim 20\%$ (compared to the maximum $\Delta S_{it(q-d)}$ at this pressure) due to hysteresis losses. In any case the reversible entropy change is still very high, $\Delta S_{it(rev)} = 30.5$ J kg$^{-1}$ K$^{-1}$.

On the other hand for higher pressures ($P \geq 500$ bar), under which the transition temperature shift is already one order of magnitude higher than thermal hysteresis, the reversible entropy change is $\Delta S_{it(rev)} = 32.9$ J kg$^{-1}$ K$^{-1}$ ($P = 500$ bar), and $\Delta S_{it(rev)} = 26.5$ J kg$^{-1}$ K$^{-1}$ ($P = 1,000$ bar), and the temperature range for reversibility gets considerable broaden.

These results in turn imply that for practical applications the best working conditions to obtain an optimized caloric

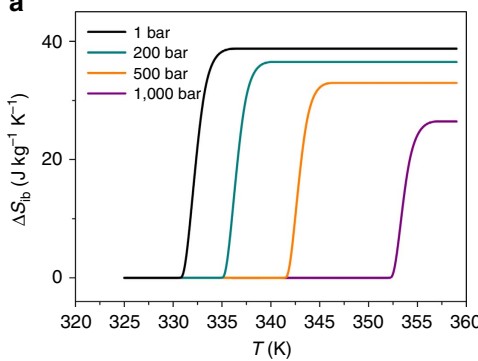

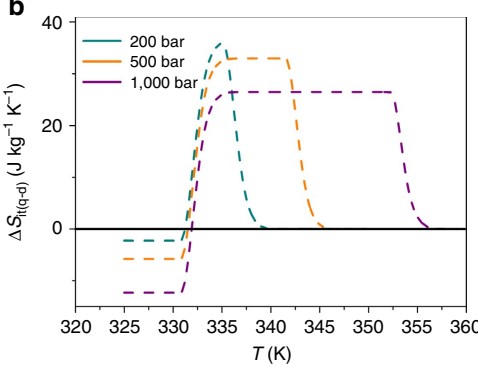

**Figure 3 | Barocaloric effect at higher pressures.** (**a**) Isobaric entropy change as a function of temperature in the higher pressure range (1–1,000 bar) on heating. (**b**) Barocaloric effect calculated as the difference between $\Delta S_{ib}$ curves obtained at different pressures by this quasi-direct method. It should be noted that these $\Delta S_{it(q-d)}$ curves have been offset at each pressure using additional entropy change $\Delta S_+(P)$ at $T_+ = 358$ K and that $\Delta S_+(P)$ has been evaluated at such $T_+$, which is above and far from the phase transition temperature, where the sample is always in the high temperature polymorph II.

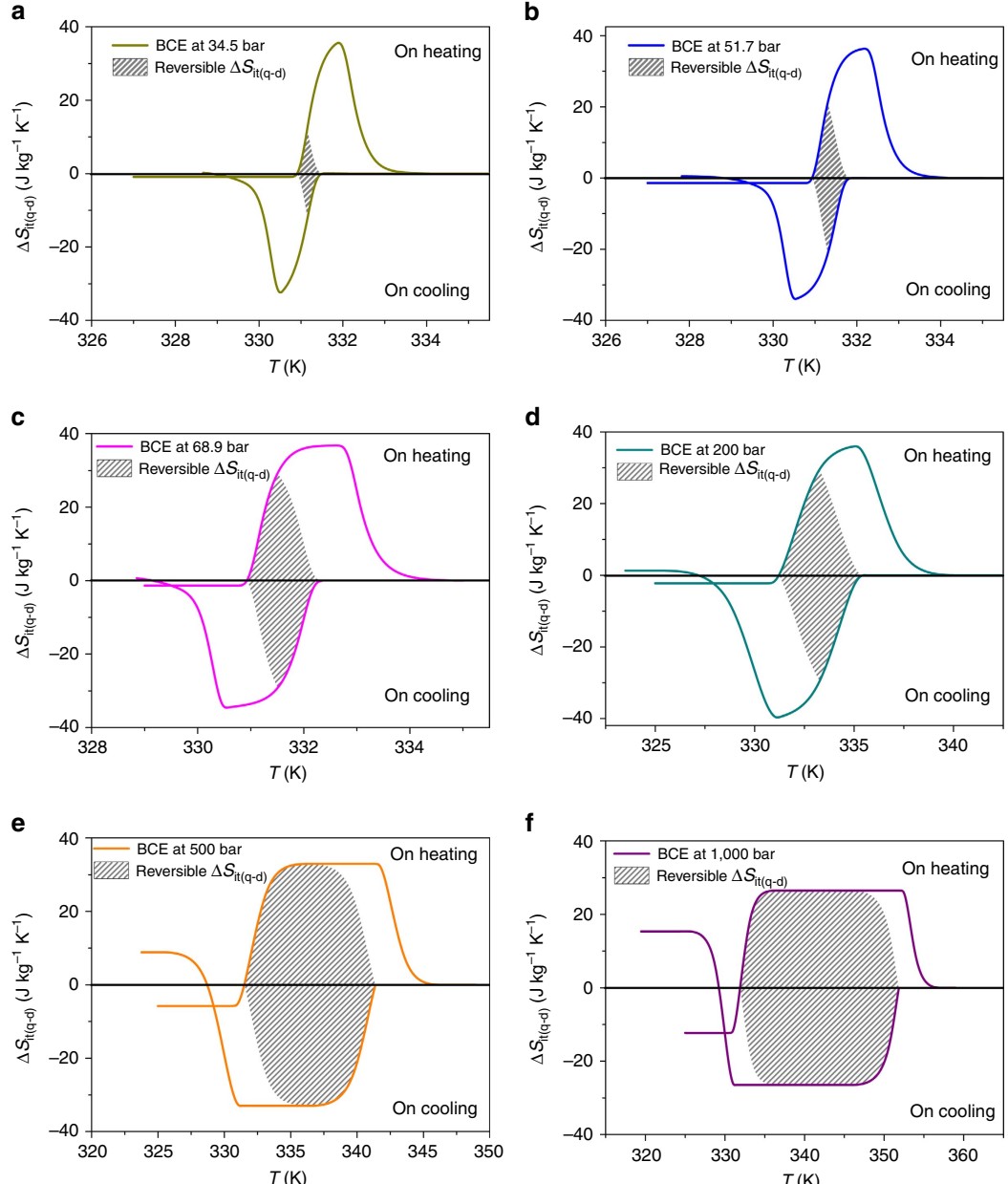

**Figure 4 | Reversible entropy change.** Isothermal entropy change calculated by quasi-direct methods on heating and cooling for pressure values from 34.5 bar to 1,000 bar (**a**–**f**). These $\Delta S_{\text{it}(q-d)}$ curves have been offset at each pressure using additional entropy change $\Delta S_+(P)$ at $T_+ = 358$ K. The grey shaded area represents the reversible entropy changes at each pressure.

performance would be those which, using the lower possible pressures, allow to obtain a strong enough barocaloric effect and a good compromise among hysteresis losses, the pressure-induced decrease of entropy, and the temperature interval where the caloric effect remains reversible.

Moreover, we expect that such caloric performance can be further improved and optimized in future studies by modifying intrinsic factors (by chemical doping, for example) and/or extrinsic factors such as the sample's microstructure as it is currently being done in the investigations on the related magnetocaloric materials[14].

**Isothermal entropy change by direct methods.** Direct measurements of entropy change can be challenging and therefore, there are few works reporting this type of analysis[4].

In this section we present the results of isothermal entropy change by direct analysis obtained on the P-DSC equipment under the cyclic application/release of a load of 68.9 bar to the [TPrA][Mn(dca)$_3$] sample. Taking into account that the phase transition of this material shifts towards higher temperatures when applying pressure, to carry out such direct isothermal calorimetric measurements we have used the following protocol: (i) the sample is heated up to 331.4 K in order to stabilize the high temperature polymorph II. (ii) Once this temperature is set and the phase transition has occurred, we perform several cycles by applying/removing a pressure of 68.9 bar in isothermal conditions (see more details in Methods). Such protocol gives rise to a pressure-driven structural transition from polymorph II to polymorph I, and vice versa, as evinced by exo- and endothermic peaks (see Fig. 5). The so-obtained entropy change exhibits a value of $\Delta S_{\text{it}(\text{direct})} = 35.1$ J kg$^{-1}$ K$^{-1}$,

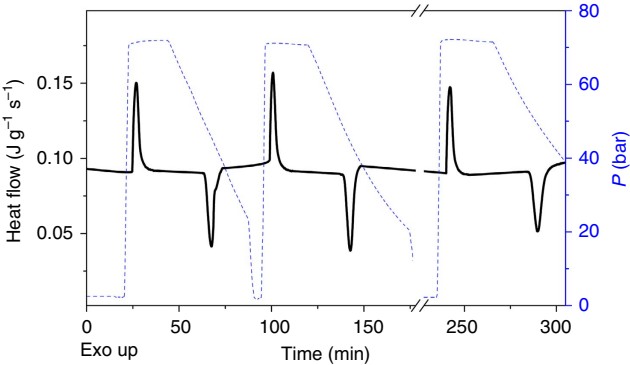

**Figure 5 | Direct isothermal entropy change.** Calorimetric curves (black solid line) obtained in isothermal conditions under cyclic application/removal of external hydrostatic pressure (blue dash line).

which is reproducible and is fully in agreement with the values obtained by quasi-direct methods. It should be noted that by means of this direct method we obtain a maximized reversible entropy change where the hysteresis losses seem to be minimized, probably due to kinetic factors related to the rate of pressure application/removal.

**Indirect Clausius–Clapeyron method.** Since this material undergoes a first-order phase transition, the entropy change associated with the barocaloric effect can be also calculated using the indirect Clausius–Clapeyron method and equation (3) (ref. 4):

$$(\delta T_t/\delta P) = \Delta v/\Delta S_{C.C.} \tag{3}$$

where $\delta T_t/\delta P$ is the transition temperature dependence with the applied pressure, also known as the barocaloric coefficient ($\delta T_t/\delta P = 23.1\,\mathrm{K\,kbar^{-1}}$); $\Delta v$ is the change in the specific volume of the material, which we have obtained from previous synchrotron PXRD data[36] ($\Delta v \sim 1 \times 10^{-5}\,\mathrm{m^3\,kg^{-1}}$); and $\Delta S_{C.C.}$ is the isothermal entropy change calculated by means of this Clausius–Clapeyron method. The obtained result, $\Delta S_{C.C.} = 43.2\,\mathrm{J\,kg^{-1}\,K^{-1}}$, is in fully agreement with that obtained by quasi-direct and direct calorimetric measurements, $\Delta S_{it(q-d)} = 37.0\,\mathrm{J\,kg^{-1}\,K^{-1}}$ and $\Delta S_{it(direct)} = 35.1\,\mathrm{J\,kg^{-1}\,K^{-1}}$.

**Relative cooling power.** To compare the refrigeration capacity of this hybrid with that of other caloric materials, we calculate its relative cooling power (RCP), a figure of merit used to assess the usefulness of a given material and defined as shown in equation (4) (refs 21,41):

$$\mathrm{RCP} = -\Delta S_{it}^{max}\,\delta T_{FWHM}, \tag{4}$$

where $\delta T_{FWHM}$ is the full width at half maximum for the maximum isothermal entropy change, $\Delta S_{it}^{max}$. It should be indicated that for the same magnitude of applied-stimulus (pressure, magnetic or electric field), a larger RCP indicates a larger refrigerating capacity and thus a better caloric material. The RCP of [TPrA][Mn(dca)₃] at 68.9 bar (a relatively low pressure) turns to be as high as $\sim 66\,\mathrm{J\,kg^{-1}}$, as indicated in Table 1, where selected caloric parameters for the up to now best baro-, magneto- and electrocaloric materials are summarized and compared. In addition to the isothermal entropy change ($\Delta S_{it}$) and adiabatic temperature change ($\Delta T_{ad}$) for each material, Table 1 also includes the value of the external stimulus applied to the caloric material (namely, value of pressure-$|\Delta P|$-, magnetic field -$|\Delta H|$- or electric field -$|\Delta E|$-), the maximum RCP achieved under those conditions, and the RCP normalized per magnitude

of the applied stimulus (RCP/$|\Delta P|$, RCP/$|\Delta H|$, RCP/$|\Delta E|$). Interestingly enough, and as it can be seen there, the RCP of [TPrA][Mn(dca)₃] per pressure unit is very competitive with the best barocaloric materials[8,19–21,42–44], and widely overpass the RCP per magnetic or electric field unit of the best performing magneto- and electrocaloric materials[9,21,45].

**Discussion**

In this work, we present an organic–inorganic hybrid compound showing a giant barocaloric effect, even under very low applied pressures. It is the hybrid [TPrA][Mn(dca)₃] perovskite material, in which such a unique behaviour is related to a complex structural solid–solid phase transition at $T_t \sim 330\,\mathrm{K}$ driven by the synergistic association of the off-centre displacements of the TPrA cations and order–disorder of their pending propyl groups, as well as the [MnN₆] octahedral tilting and order–disorder associated with the dca bridging ligands[36].

Such barocaloric effect has been characterized by calorimetry under constant pressure (on heating and cooling) and under the cyclic application/release of a load using quasi-direct and direct entropy methods.

Very remarkably, the isothermal application of pressure to this hybrid compound with a barocaloric coefficient ($\delta T_t/\delta P = 23.1\,\mathrm{K\,kbar^{-1}}$), results in a very large caloric effect of $\Delta S_{it(q-d)} = 37.0\,\mathrm{J\,kg^{-1}\,K^{-1}}$, slightly above room temperature, and an adiabatic temperature change $\Delta T_{ad(q-d)} \sim 5.0\,\mathrm{K}$ already under very small applied pressures ($P = 68.9\,\mathrm{bar}$).

Also, as the heating and cooling runs reveal the reversible entropy change is rather large, $\sim 31.0\,\mathrm{J\,kg^{-1}\,K^{-1}}$, even under those small pressures, although the use of larger pressures results in a broader temperature span of applicability.

Very interestingly, the fact that in this hybrid compound the entropy change is mainly determined by the entropy of some atomic arrangements that get partially ordered/disordered implies that there is plenty of room to get higher values: not only by improving the configurational entropy, but also by playing with rotational and vibrational entropies, which at difference with the case of inorganic perovskites can be also relevant in organic–inorganic hybrid perovskites[46–49].

Therefore, we expect that the here reported findings will lead to follow-on efforts for tuning, doping and tailoring the molecular building blocks of this and related systems, as well as the microstructure of the samples, to obtain barocaloric materials with enhanced configurational, rotational and vibrational entropic effects and optimized caloric performance.

In conclusion, this work opens up a new and exciting horizon for the scientific community working on materials science, in general, and in organic–inorganic hybrids, in particular, as many of this large, flexible, multifunctional and designable class of compounds exhibit the basic ingredients to also display large caloric effects, namely: large and reversible entropy changes commonly related to phase transitions of order–disorder nature[50,51] (often associated, in turn, with ferroic transitions), as well as remarkable external multi-stimuli responses (towards magnetic and/or electric field, pressure, temperature, and so on), which can be especially strong in the case of applying pressure due to their rather high flexible structures. As a result, it also offers new interesting opportunities for the development of cooling devices based on environmentally friendly and economically more accessible solid-state materials.

**Methods**

**Synthesis.** Single phase polycrystalline [TPrA][Mn(dca)₃] was prepared from aqueous solution at ambient temperature as reported[36]. All reagents were commercially available and were used as purchased without further purification: Mn(NO₃)₂·xH₂O (97%, Aldrich), (TPrA)Br (98%, Aldrich), Na(dca)

(96%, Aldrich), and absolute ethanol (Panreac). Also, a reagent amount of deionized water was used in the synthesis. A mixture of (TPrA)Br (2 mmol) solution in ethanol (10 ml) and a solution of Na(dca) (6 mmol) in water (10 ml) was layered on top of an aqueous solution of $Mn(NO_3)_2 \cdot xH_2O$ (2 mmol). Colourless polycrystalline powder was obtained and collected by filtration and washed several times with ethanol.

**Powder X-ray diffraction.** A Siemens D-5000 diffractometer, using Cu $K_\alpha$ radiation ($\lambda = 1.5418$ Å), was used to study this compound by PXRD at room temperature. The PXRD pattern was compared with that reported in the literature[36]. The purity of the as-synthesized compound, which is a single phase material, was confirmed by Le Bail refinement of PXRD patterns. PXRD analysis was also performed after sample compression/decompression cycles, results which showed that the material is still fully crystalline and polymorph I is the phase present at room conditions (see Supplementary Fig. 4).

**Calorimetry in the low pressure region.** Differential scanning calorimetry analysis from 1 bar to 68.9 bar were carried out in a TA Instruments Q2000 modulated DSC equipped with a pressure cell. Indium was used as standard for temperature and heat calibration of the cell at each pressure[52,53]. Nitrogen gas was used to maintain a constant pressure (from 1 to 68.9 bar) on the sample ($\sim 5$ mg of [TPrA][Mn(dca)$_3$] confined inside a pinhole aluminium capsule), keeping a constant gas flow of 50 ml min$^{-1}$. Heating and cooling ramps at 2 K min$^{-1}$ were programmed from room temperature to 343 K. However, as a consequence of the intrinsic features of pressure DSC cells, temperature control during the cooling ramp is worse than on heating and some deviations were observed. Therefore, for low pressures, barocaloric effects calculated from heating data are more precise than those of cooling data. Several cycles of pressure application/release were performed in the sample at each pressure value, confirming the reversible behaviour of the process and its reproducibility. Moreover, the same behaviour was observed in different batches.

Same conditions were used for the cyclic isothermal P-DSC analysis. These experiments were carried out under the repeated application/removal of a hydrostatic pressure of 68.9 bar at $T = 331.4$ K, selected temperature which was maintained constant by the calorimeter all along the experiment.

Owing to the equipment limitation in controlling the rate of pressure application/release, such pressure was applied at a rate of 30 bar min$^{-1}$ and released at a rate of 1 bar min$^{-1}$. In any case, it should be noted that the difference between both rates did not imply any difference in the obtained values of entropy change on applying and/or removing the pressure and also that the isothermal conditions were maintained all along the experiments.

**Calorimetry in the higher pressure region.** P-DSC analysis from 1 bar to 1,000 bar were performed in a Setaram μDSC7 EVO equipment. Around 130 mg of sample were introduced in a high pressure (HP) vessel that was inserted in the measurement well of the calorimeter, previously equilibrated at 25 °C. An empty HP vessel was inserted in the reference well of the calorimeter in order to compensate the heat capacity of the vessel. The two HP cells were tightly closed and were connected with a panel gas ISCO pump. Heating and cooling ramps were performed in similar conditions that in the low pressure region from room temperature to 370 K. Temperature and heat calibration was also performed at each experimental condition.

**Estimation of additional entropy changes $\Delta S_+(P)$.** This isothermal entropy $\Delta S_+(P)$ was estimated away from the first-order transition by using the equation $\Delta S_+(P) = -[m^{-1}(\delta V/\delta T)_{P=0}]P$ (ref. 54), where $m$ is the mass, $P$ is the applied pressure and $(\delta V/\delta T)_{P=0}$ was obtained from temperature dependence of the unit cell volume at ambient pressure. The values of unit cell volume have been obtained from Rietveld analysis of synchrotron PXRD at different temperatures (see details in ref. 36).

**Specific heat capacity.** The heat capacity measurements were performed in a TA Instruments MDSC Q2000 with standard (non-pressure) cell in quasi-isothermal modulated mode, at room pressure and at 55 °C (328 K) temperature that, although near, is clearly separated from the transition at ambient pressure. The modulation parameters were an amplitude of $\pm 0.5$ K and a period of 60 s.

**Data availability.** All relevant data are presented via this publication and Supplementary Information.

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

## Acknowledgements

The authors are grateful for the financial support from Ministerio de Economía y Competitividad MINECO and EU-FEDER under project ENE2014-56237-C4-4-R and Xunta de Galicia under project GRC2014/042. J.M.B.-G. wants to thanks Fundación Barrié for a predoctoral fellowship.

## Author contributions

M.A.S.-R. conceived the study and led the project. M.A.S.-R., J.M.B.-G., M.S.-A., S.C.-G., J.L.-B. and R.A. planned the experiments. J.M.B.-G. and M.S.-A. synthesized the compound under study. S.C.-G., J.L.-B. and R.A. performed the calorimetric measurements at atmospheric pressure. J.M.B.-G., J.L.-B. and R.A. performed the calorimetric measurements under pressure. M.S.-A. and S.C.-G. performed the X-ray diffraction measurements. M.A.S.-R. and J.M.B.-G. wrote the manuscript with input from M.S.-A., S.C.-G., J.L.-B. and R.A.

## Additional information

**Competing interests:** The authors declare no competing financial interests.

