## [Peer Review File · Nature Communications]

Reviewers' comments:

Reviewer #1 (Remarks to the Author):

Bermúdez-García et al. discuss about very unique results on the barocaloric effect of organic-inorganic compound. The striking feature is that this material exhibit a caloric change comparable to other systems under tremendously small pressure (0.00689 GPa) around room temperature.

Although the authors mentioned that this barocaloric effect comes from the first order phase (polymorph) transition, the microscopic physical origin of entropy (or latent heat) is not easy to understand from their explanations. What is the factor which determines the magnitude of the latent heat; entropy of atomic arrangements? any charge or spin fluctuations? etc. etc. (This point restricts the maximum value of caloric change, therefore, the physical origin may give answer to what extent the magnitude of this type of caloric effect WILL go beyond by materials tuning. At the present stage, its strength per pressure is largest, but its magnitude is easily saturated by very small pressures.)

Reviewer #2

"Giant barocaloric effect in the ferroic organic-inorganic hybrid [TPrA][Mn(dca)₃] perovskite under easily accessible pressures" by Prof Señarís-Rodríguez et al.

In this paper the authors present experimental data of the barocaloric effect in a ferroic organic-inorganic hybrid [TPrA][Mn(dca)₃] perovskite. They show that this material exhibits giant barocaloric effect at low pressures (lower than 70 bar).

The experimental data are interesting but they seem to be quite similar to the previous work published by the authors (ref 35 of the manuscript).

There are some points to be addressed by the authors before the final decision. I have several concerns and doubts about the correctness of the interpretation of the experimental results, concerns that should be clarified before I can recommend the publication of the paper. The following lines detail my concerns, doubts and suggestions of improvement.

* It has profusely been demonstrated that the porous structure of metal-organic frameworks (MOFs) makes quite easy the adsorption of gases. Even for organic materials with higher packing it has been shown that application of pressure by means of a gas can modify the phase transitions as well as its pressure dependence (the slope dT/dP). The

authors should verify that gas (nitrogen in this case) has no influence on the properties of the phase transition, so calorimetric measurements on encapsulated samples (i.e., isolated from the gas used for increase the pressure) must be performed.

*Many of the data here presented were previously published in ref. 35 of the manuscript, from the same group. The data there presented concern:

a) Calorimetric measurements: The next figures appear to be quite similar, the left figure (from ref. 35) provides 4 thermograms as a function of pressure, whereas the right figure (from the present manuscript) displays 5 calorimetric curves. As the sample is known to be the same, the authors should explicitly mention previous measurements. Moreover, the inset of the figure 2 (supplementary information) should appear at the main manuscript, insomuch as authors claim as one of the more relevant results. Even more, if ref. 35 provided data of the phase transition as a function of pressure, why these experimental points are not added to the pressure-temperature phase diagram here presented? Such extension would improve the error of the slope (on heating) $dT/dP=25.6 \text{ K kbar}^{-1}$ (or $24.1 \pm 0.1 \text{ K kbar}^{-1}$ according to ref.35).

b)

a) The entropy change associated with the transition is here given as $45 \text{ J kg}^{-1} \text{ K}^{-1}$. Nevertheless, in ref. 35, from the authors, the enthalpy change for the same compound and the same transition is given as 6077 J mol^{-1} , which provides an entropy change of $41.9 \text{ J kg}^{-1} \text{ K}^{-1}$, quite different from the value quoted in this manuscript. Such a disagreement deserves a comment.

b) The pressure range for the calorimetric measurements is really narrow (0-70 bar). In fact, it is so narrow that the change of the temperature within such a domain is of the order of the thermal hysteresis (line 95 of the manuscript). To account for such a problem the authors should: (i) perform additional calorimetric measurements at higher pressure and (ii) calculate also the isobaric entropy change also from cooling measurements. The authors comment on the “reversibility of the transition” (line 99 of the manuscript), but I guess that authors are confused with the differences in the entropy change on isothermally increasing and decreasing pressure (which would evidence, if calculated) the irreversible barocaloric effects.

c) In figure 1 (*Isobaric entropy change*) the slope of the entropy difference at the low-temperature phase (polymorph I) is clearly different from 0. There is only one possibility for such a result: the authors have taken into the specific heat of this phase and thus the origin of entropy is quite unclear. This then implies that the authors have measured the specific heat or its values are coming from literature. None of these options are described in the text. Only in line 162 the authors wrote “...estimated in this case to be $2450 \text{ J K}^{-1} \text{ kg}^{-1}$...”, but no reference is cited. On the contrary, the slope of the entropy difference within the temperature range of polymorph II (high-temperature region) seems to be zero, which would mean that there the specific heat was not taken into account. The lack of details makes it difficult to understand the manuscript.

*Volume change at the transition and elastic contribution: The volume as a function of temperature (at normal pressure) and calculation of the isobaric thermal expansion parameter were obtained from previous measurements published in ref. 35. Despite that this is not very clear in the manuscript, authors should provide such information (at least volume as a function of temperature) in the Supp. Information. The reason why this reviewer is asking for such data is that I disagree with the method used for determining the “volumetric thermal expansion coefficient” (β_V), which appears as a figure into the Supp Information of reference 35.

In light of the previous comments, I do not recommend this manuscript for publication in Nature Comm.

Reviewer #3 (Remarks to the Author):

In this manuscript, Bermudez-Garcia et al. put forward hybrid perovskites for barocaloric applications. These materials are potentially very interesting but additional work is needed to demonstrate the barocaloric performance of these compounds, as explained below.

Due to the thermal hysteresis of the transition, application of low pressures yield irreversible entropy changes (and temperature changes) that cannot be used for cooling applications (as seen in figure S2). Authors should therefore perform calorimetric measurements at higher hydrostatic pressures in order to demonstrate reversible entropy changes (and temperature changes) that can be used for cooling applications.

Recent publications on barocaloric effects have shown how the detrimental impact of hysteresis on the barocaloric response can be overcome using sufficiently high pressures in order to yield reversible barocaloric effects. Authors should examine these previous publications carefully when revising their work.

Reviewer #1:

Bermúdez-García et al. discuss about very unique results on the barocaloric effect of organic-inorganic compound. The striking feature is that this material exhibit a caloric change comparable to other systems under tremendously small pressure (0.00689 GPa) around room temperature.

Although the authors mentioned that this barocaloric effect comes from the first order phase (polymorph) transition, the microscopic physical origin of entropy (or latent heat) is not easy to understand from their explanations. What is the factor which determines the magnitude of the latent heat; entropy of atomic arrangements? any charge or spin fluctuations? etc. etc. (This point restricts the maximum value of caloric change, therefore, the physical origin may give answer to what extent the magnitude of this type of caloric effect WILL go beyond by materials tuning. At the present stage, its strength per pressure is largest, but its magnitude is easily saturated by very small pressures.)

In this compound, the microscopic origin of entropy is related to changes in atomic arrangements at the phase transition. In this context, we have previously reported a detailed study of the structural phase transition that takes place between two different polymorphs (*polymorph I* and *II* (ref. 36: *Inorg. Chem.* **54**, 11680-11687 (2015))). As explained there in depth, this structural phase transition mainly involves an 8-fold order-disorder process of the N and C atoms of the dca ligands and the propyl groups of the TPrA cations in the crystal structure.

Following this comment, we have clarified the origin of the entropy change in the revised manuscript (**page 6**: “As referred above, and explained in depth in a previous publication,³⁶ the origin of this entropy increase is related to an structural order-disorder phase transition in this crystalline compound, involving mainly the atoms of the dca ligands and the propyl groups of the TPrA cations.”).

On the other hand, and as this reviewer remarks, the entropy associated with such process exhibits a maximum value that is easily saturated at small pressures, which is one of the advantages of this system.

In that regard, it should be highlighted that in comparison with oxides or alloys, organic-inorganic hybrid compounds are relatively flexible materials where some atoms often display a larger degree of freedom.

From the combination of these two characteristics, this phase transition involves a large entropy change and a high sensibility towards external pressure, so that the maximum value of caloric change is already reached at small pressures, which can be very advantageous for practical applications.

Finally, and answering to the last part of the question raised by this reviewer, it will be possible indeed to increase this maximum entropy by tuning or doping the material, looking for increasing the order of the low temperature *polymorph I* and the disorder of the high temperature *polymorph II*. Consequently, this work not only reveals an unprecedented barocaloric response in a organic-inorganic hybrid compound, but also encourages future studies on tuning, doping and tailoring the molecular building blocks of this type of systems, as well as the microstructure of the prepared samples, to get an optimized caloric response.

Reviewer #2:

"Giant barocaloric effect in the ferroic organic-inorganic hybrid [TPrA][Mn(dca)₃] perovskite under easily accessible pressures" by Prof Señarís-Rodríguez et al. In this paper the authors present experimental data of the barocaloric effect in a ferroic organic-inorganic hybrid [TPrA][Mn(dca)₃] perovskite. They show that this material exhibits giant barocaloric effect at low pressures (lower than 70 bar).

The experimental data are interesting but they seem to be quite similar to the previous work published by the authors (ref 35 of the manuscript).

There are some points to be addressed by the authors before the final decision. I have several concerns and doubts about the correctness of the interpretation of the experimental results, concerns that should be clarified before I can recommend the publication of the paper. The following lines detail my concerns, doubts and suggestions of improvement.

1. It has profusely been demonstrated that the porous structure of metalorganic frameworks (MOFs) makes quite easy the adsorption of gases. Even for organic materials with higher packing it has been shown that application of pressure by means of a gas can modify the phase transitions as well as its pressure dependence (the slope dT/dP). The authors should verify that gas (nitrogen in this case) has no influence on the properties of the phase transition, so calorimetric measurements on encapsulated samples (i.e., isolated from the gas used for increase the pressure) must be performed.

As reviewer 2 remarks, it is well-known that porous MOFs and porous coordination polymers (PCPs) have very "open" structures that account for their remarkable capability to adsorb gases into their nanoporous cavities. But it should also be indicated that, despite their "open" structures, even such compounds must be in any case previously activated to reach their adsorption properties: typically by heating them under low pressure conditions so as to previously remove from their cavities the solvent, guest species or moisture molecules allocated there to make room for the new molecules one wants to adsorb.

On the other hand, it is also well-known that there is another group of MOFs and coordination polymers that present a much denser crystalline structure, where the gas adsorption is negligible, but that are very interesting in view of the functional properties they can display: magnetic, electric, etc. (**ref. 24:** Cheetham, A. K. & Rao, C. N. R. There's room in the middle. *Science* **318**, 58-59 (2007)).

In these "dense" hybrids, the cavities left by the framework are much smaller and typically, one cation is located inside. Such cation is linked to the framework by different types of chemical interactions and cannot be removed without collapsing of the crystal structure. Therefore, there is no room for other guest molecules and these materials cannot adsorb gases.

The here presented compound belongs to this second group of "dense" hybrids. In any case, following this comment, we have checked its crystal structure with the PLATON software in order to search for voids where small atoms like He could be located, but as we expected the crystal structure is so dense that there is no space for them.

To visualize the volume of the cavities, Figure L1 shows the Hirshfeld surface

of the TPrA cation inside the pseudo-cubooctahedral cavity of the $[\text{Mn}(\text{dca})_3]^-$ framework.

As it can be seen, the TPrA cation fills almost completely all the cavity volume and the voids are negligible, even if the hydrogen atoms of the TPrA cation (28 H-atoms) have not been taken into account for this calculation, and are not included in the picture.

Figure L1. Calculated Hirshfeld surface of the TPrA cation inside the pseudo-cubooctahedral cavity of the $[\text{Mn}(\text{dca})_3]^-$ framework. It should be noted that the hydrogen atoms of the TPrA cation have not been taken into account for this calculation.

Moreover, our previous experimental results (**ref. 36: *Inorg. Chem.* 54, 11680-11687 (2015)**, former ref. 35 in the previous version of the manuscript) have profusely proven that the phase transition and the $\delta T_i/\delta P$ dependence of this compound exhibit the same behavior in different atmospheres and environments: under nitrogen gas (in the Pressure DSC analysis), under silicon oil (in the dielectric measurements under pressure) and under room atmosphere (under mechanical uniaxial pressure in a stain anvil during PVT analysis).

Therefore, we can undoubtedly conclude that the very remote possibility of some nitrogen adsorption is not a factor influencing neither the phase transition nor the $\delta T_i/\delta P$ dependence of this compound.

Thus, we consider than additional calorimetric measurements on encapsulated samples are not necessary. Besides such measurements, which are very tricky, can be easily out of isobaric conditions due to the increase of pressure inside the container when heating it up.

2.a) Many of the data here presented were previously published in ref. 35 of the manuscript, from the same group. The data there presented concern:

a) Calorimetric measurements: The next figures appear to be quite similar, the left figure (from ref. 35) provides 4 thermograms as a function of pressure, whereas the right figure (from the present manuscript) displays 5 calorimetric curves. As the sample is known to be the same, the authors should explicitly mention previous measurements. Moreover, the inset of the figure 2 (supplementary information) should appear at the main manuscript, insomuch as authors claim as one of the more relevant results. Even more, if ref. 35 provided data of the phase transition as a function of pressure, why these experimental points are not added to the pressure-temperature phase diagram here presented? Such extension would improve the error of the slope (on heating) $dT/dP=25.6$ K kbar⁻¹ (or 24.1 ± 0.1 K kbar⁻¹ according to ref.35).

Following this reviewer's comment, we have moved the inset from Figure 2, Supplementary Information, to the main manuscript as Figure 1. We have also added new points to this Figure 1, including those corresponding to measurements done at higher pressures (up to 1000 bar) and both on heating and cooling runs. Taking into account the data for that whole pressure range, we obtain a value $\delta T_i/\delta P$ of 23.1 K kbar⁻¹, which is slightly smaller than that previously calculated from a smaller pressure range but consistent with the data obtained by other techniques in broader pressure ranges.

And we explicitly mention our previous measurements of $\delta T_i/\delta P$ in the revised manuscript (**page 5**: "From these data, we have calculated the so-called barocaloric coefficient ($\delta T_i/\delta P$) as the variation of the transition temperature as a function of pressure obtaining a value of $\delta T_i/\delta P = 23.1$ K kbar⁻¹ (see Figure 1). This value is in good agreement with our previous results obtained by different techniques (P-DSC, dielectric measurements under pressure and pressure-volume-temperature -PVT-analysis).³⁶").

2.b) The entropy change associated with the transition is here given as 45 J kg⁻¹ K⁻¹. Nevertheless, in ref. 35, from the authors, the enthalpy change for the same compound and the same transition is given as 6077 J mol⁻¹, which provides an entropy change of 41.9 J kg⁻¹ K⁻¹, quite different from the value

quoted in this manuscript. Such a disagreement deserves a comment.

Following this comment, we have rechecked very carefully the value of the entropy change by performing several measurements on several batches of samples, and also by using different instruments and amount of sample. In this context, we have found that the most accurate value of the entropy change of the phase transition is $42.5 \text{ J kg}^{-1} \text{ K}^{-1}$. It should be noted that this very large entropy change value is very similar to that given in the previous version of the manuscript and to the value reported in **ref. 36** (former **ref. 35** in the previous version of the manuscript: *Inorg. Chem.* **54**, 11680-11687 (2015)), and that the small difference among them can be attributed to experimental errors.

The recalculated entropy change has been included in the new version of the manuscript (**page 4**: “*The obtained results fully corroborate, in first place, the first-order nature of the phase transition [...] and a large entropy change of $\Delta S_{\text{phase}} \sim 42.5 \text{ J kg}^{-1} \text{ K}^{-1}$, see Supplementary Figure 1.*”)

2.c) The pressure range for the calorimetric measurements is really narrow (0-70 bar). In fact, it is so narrow that the change of the temperature within such a domain is of the order of the thermal hysteresis (line 95 of the manuscript).

To account for such a problem the authors should: (i) perform additional calorimetric measurements at higher pressure and (ii) calculate also the isobaric entropy change also from cooling measurements. The authors comment on the “reversibility of the transition” (line 99 of the manuscript), but I guess that authors are confused with the differences in the entropy change on isothermally increasing and decreasing pressure (which would evidence, if calculated) the irreversible barocaloric effects.

We very much thank these criticisms, which have helped us to significantly improve the manuscript.

The reviewer is right in saying that the pressure range is very narrow and of the order of thermal hysteresis.

Addressing such problem, we have found that the thermal hysteresis can be reduced down to 0.9 K by using a heating/cooling rate of 2 K min^{-1} (**page 4**: “*The obtained results fully corroborate, in first place, the first-order nature of the phase transition, which displays a small hysteresis of $\sim 0.9 \text{ K}$ at a heating/cooling rate of 2 K min^{-1} [...] see Supplementary Figure 1.*”)

Moreover, in order to demonstrate the reversible entropy change and following the recommendation of this reviewer, (i) we have performed additional calorimetric measurements at higher pressures (up to 1000 bar) so as to be able to reach a temperature change by application of external pressure which is more than one order of magnitude larger than thermal hysteresis, see Figure L2. And (ii) we have calculated the isobaric entropy change also from cooling measurements, see Figure L2, where the grey shaded area represents the reversible entropy change.

All these new results have been presented and discussed in the revised version of the manuscript (from **page 9 to 11**), where we have included these new figures as Figure 4 and 5, and Supplementary Information Figures 5 and 6.

Figure L2. Reversible entropy change. Isothermal entropy change calculated by quasi-direct methods on heating and cooling for pressure values from 34.5 bar to 1000 bar. These curves have been normalized respect to the temperature interval above the transition to facilitate the view of the reversible region of the entropy change, which is represented by the grey shaded area. (Supplementary Figure 6 of the Supplementary Information).

In addition to these measurements, we have also carried out additional new direct entropy change measurements, which, as can be seen in Figure L3, show a very good reproducibility with reversible isothermal entropy change ($\Delta S_{it(\text{direct})} = 35.1 \text{ J kg}^{-1} \text{K}^{-1}$). These new results have been included and discussed in the revised

version (from **page 11 to 12**, and as a new Figure 6).

Figure L3. Direct isothermal entropy change. Calorimetric curves (black solid line) obtained in isothermal conditions under cyclic application/removal of external hydrostatic pressure (blue dash line). (Figure 6 of the Manuscript).

Also, following the comments of this reviewer, we have clarified in the revised version of the manuscript what we mean with “reversibility of the transition” (**page 5**: “*Very interestingly, the initial situation is recovered after decompression, even when performing several cycles with the material showing a reproducible pressure cyclability, see Supplementary Figure 2. The reversibility of this process is also corroborated by powder X-ray diffraction data, which show that after decompression (from 68.9 bar and 1000 bar, respectively) the material is still fully crystalline and polymorph I is the phase present at ambient conditions, see Supplementary Figure 3.*”).

2.d) In figure 1 (Isobaric entropy change) the slope of the entropy difference at the low-temperature phase (polymorph I) is clearly different from 0. There is only one possibility for such a result: the authors have taken into the specific heat of this phase and thus the origin of entropy is quite unclear. This then implies that the authors have measured the specific heat or its values are coming from literature. None of these options are described in the text. Only in line 162 the authors wrote “...estimated in this case to be 2450 J K⁻¹ kg⁻¹...”, but not reference is cited. On the contrary, the slope of the entropy difference within the temperature range of polymorph II (high-temperature region) seems to be zero, which would mean that there the specific heat was not taken into account. The lack of details makes it difficult to understand the manuscript.

Following the recommendations of the reviewer, we have added details about the technique used to estimate the specific heat in the “Methods” section of the revised manuscript (**page 18**: “**Specific heat capacity.** *The heat capacity measurements were performed in a TA Instruments MDSC Q2000 with standard (non pressure) cell in quasi-isothermal modulated mode, at room pressure and at 55 °C (328 K) temperature that, although near, is clearly separated from the transition at ambient pressure. The modulation parameters were an amplitude of ± 0.5 K and a period of 60 s.*”).

The reviewer is right and we had forgotten to include it in the previous version of the manuscript.

In addition, following these comments, we have also revised the former Figure 1, and we have replotted it as Figure 2 in the revised manuscript.

***Volume change at the transition and elastic contribution: The volume as a function of temperature (at normal pressure) and calculation of the isobaric thermal expansion parameter were obtained from previous measurements published in ref. 35. Despite that this is not very clear in the manuscript, authors should provide such information (at least volume as a function of temperature) in the Supp. Information. The reason why this reviewer is asking for such data is that I disagree with the method used for determining the “volumetric thermal expansion coefficient” (β_v), which appears as a figure into the Supp Information of reference 35.**

Following this comment, in the revised manuscript we have clarified that the thermal expansion parameter was obtained from previous data published in **ref. 36**, former **ref. 35** (see **page 18**: “**Elastic heating calculation.** *The elastic heating ($\sim \beta \times v \times \Delta P$) was calculated from the volumetric thermal expansion coefficient (β) and the specific volume (v), see Supplementary Figure 7, both obtained from previously reported data of synchrotron powder X-ray diffraction³⁶”).*

As for how we had calculated in reference 36 (former **ref. 35**) the volumetric thermal expansion coefficient (β), we have to note that we had followed a method widely and commonly used in the literature [see for example J. Chen, L. Hu, J. Deng and X. Xing, *Chem. Soc. Rev.*, **44**, 3522-3567 (2015)].

In any case, in the new version of the manuscript, we have recalculated again the volumetric thermal expansion coefficient of this compound but following the method reported in **reference 8** [Lloveras, P., Stern-Taulats, E., Barrio, M., Tamarit, J-LI., Crossley, S., Li, W., Pomjakushin, V., Planes, A., Mañosa, Ll., Mathur, N. D. & Moya, X. Giant barocaloric effects at low pressure in ferroelectric ammonium sulphate. *Nat. Commun.* **6**, 8801 (2015)], see Figure L4.

These new results have been included in the revised version of the manuscript (see “Methods” section and Supplementary Figure 7).

Finally, it should be indicated that to be completely sure that the possible contribution of elastic heating to the observed effect is negligible, the elastic heating

has been calculated by using the largest value of β , in order to consider the maximum possible contribution of such elastic heating to the entropy change. Even in that most unfavourable case, the contribution from elastic heating is ~ 30 times lower than the isothermal entropy change.

Figure L4. Unit cell volume and thermal expansion. Temperature dependence of the unit cell volume (top) and volumetric thermal expansion coefficient $V^{-1}(\delta V/\delta T)$ (bottom) of *polymorph I* and *II*. (Supplementary Figure 7 of the Supplementary Information)

Reviewer #3 (Remarks to the Author):

In this manuscript, Bermudez-Garcia et al. put forward hybrid perovskites for barocaloric applications. These materials are potentially very interesting but additional work is needed to demonstrate the barocaloric performance of these compounds, as explained below.

Due to the thermal hysteresis of the transition, application of low pressures yield irreversible entropy changes (and temperature changes) that cannot be used for cooling applications (as seen in figure S2). Authors should therefore perform calorimetric measurements at higher hydrostatic pressures in order to

demonstrate reversible entropy changes (and temperature changes) that can be used for cooling applications.

Recent publications on barocaloric effects have shown how the detrimental impact of hysteresis on the barocaloric response can be overcome using sufficiently high pressures in order to yield reversible barocaloric effects. Authors should examine these previous publications carefully when revising their work.

Following the recommendation of this reviewer, we have carried out new calorimetric experiments, including those at higher pressures (up to 1000 bar) so as to be able to reach a temperature change by application of external pressure which is more than one order of magnitude larger than thermal hysteresis. And we have studied in detail the reversible entropy change and the hysteresis losses of the claimed barocaloric effect.

As commented to reviewer 2, in addition to these additional calorimetric measurements at higher pressures, we have calculated the isobaric entropy change also from cooling measurements, see the here presented Figure L2, where the grey shaded area represents the reversible entropy change. These new results have been presented and discussed in the new version of the manuscript (from **page 9** to **11**), and we have included new graphics as Figure 4 and 5, and Supplementary Figure 5 and 6.

Figure L2. Reversible entropy change. Isothermal entropy change calculated by quasi-direct methods on heating and cooling for pressure values from 34.5 bar to 1000 bar. These curves have been normalized respect to the temperature interval above the transition to facilitate the view of the reversible region of the entropy change, which is represented by the grey shaded area. (Supplementary Figure 6 of the Supplementary Information).

In addition to these new experiments, we have also carried out additional new direct entropy change measurements, which show a very good reproducibility with reversible isothermal entropy change of $35.1 \text{ J kg}^{-1} \text{K}^{-1}$, see Figure L3. These new results have been included and discussed in the revised version (from **page 11** to **12**, and as a new Figure 6).

Figure L3. Direct isothermal entropy change. Calorimetric curves (black solid line) obtained in isothermal conditions under cyclic application/removal of external hydrostatic pressure (blue dash line). (Figure 6 of the Manuscript).

Reviewers' comments:

Reviewer #1 (Remarks to the Author):

The authors courteously answer to my first question about the origin of the entropy change. Now, the readers who are not familiar this type of material can understand the mechanism of this transition.

Meanwhile, in the final paragraph, the explanation is too sloppy. In all the 1st order transition, it is no wonder that entropy change becomes large by increasing degree of order in an ordered state and degree of disorder in a disordered state; this is just a tautology.

As I mentioned in the last comment, if the authors know the mechanism of the transition, then they can estimate (an ideal) maximum entropy change, and they should compare whether the experimental result truly reach that ideal value. Then, if they believe this ideal value can be increased by such as doping and tailoring, they should explain through what mechanisms they can be modified in logical way.

This is also same comment as in the previous report; this material exhibit a largest strength per pressure, but the maximum value at the present stage is not so large (not fully enough for the practical applications)

Reviewer #2 (Remarks to the Author):

The authors have satisfactorily responded to my points of criticism and made major revisions of the manuscript following the suggestions of the referees. The authors have improved the manuscript and satisfactorily replied to most issues raised. Thus, I recommend publication of the article without further changes (there are just some typos along the Supp Information).

Reviewer #3 (Remarks to the Author):

Authors have performed the additional calorimetric measurements that were required to calculate reversible barocaloric effects.

- Figure 3, which shows entropy changes up to $38 \text{ J K}^{-1} \text{ kg}^{-1}$ that do not correspond to reversible barocaloric effects should therefore not be shown in the manuscript.

- Following the previous comment, the barocaloric response reported in table 1 should be the response associated with reversible entropy changes, which at 68.9 bar are reduced to $30 \text{ J K}^{-1} \text{ kg}^{-1}$.

- Authors should display the calorimetric data and all the subsequent analysis in the main paper, both for cooling and heating data, and for all pressures.

Inspection of the calorimetric data shown previously in figure 2 in the supplementary file reveals that entropy changes while cooling and heating at different pressures are different. These differences will modify the current data analysis, which in some cases seems too simple. Authors should therefore carefully revise their data analysis.

Also, authors should use the elastic contribution to displace the pressure-dependent entropy curves, instead of the rather arbitrary normalisation that is currently being used.

- The comparison of RCP values divided by pressure, magnetic field and electric field is meaningless, as different applied fields have different units.
- Isothermal measurements: pressure is applied at a much faster rate than it is removed. The sample will be therefore at a higher temperature while applying pressure than while removing pressure, therefore precluding isothermal conditions.

Author's response to reviewers' comments:

First of all, we thank the **reviewer 2** recommendation of publishing the article without further changes, even if he was the one that originally presented more criticisms to our work (and who in his first revision said "In light of the previous comments, I do not recommend this manuscript for publication in Nature Comm."). This reviewer did a very deep and exhaustive revision of our work, and was concerned about several aspects of the work such as the veracity of the data (possibility of nitrogen adsorption), the calculation methods used, the irreversibility of the process, the small range of pressures used, the contribution of the specific heat, the thermal expansion calculation and its contribution to the elastic heating, among others. Now this reviewer 2 agrees that we have improved the manuscript and satisfactorily replied to most issues raised. Thus, he recommends publication of the article without further changes.

We also thank **reviewer 3** for his/her new concerns, even if they had not been included in his/her previous revision. In this new revised manuscript, we have followed all his recommendations about modifying and moving figures from the manuscript to the supplementary information and vice versa, and we have also given response to the rest of his new concerns.

We also thank the **reviewer's 1** recommendation about theoretical predictions of the entropy changes and the elucidation of the transition mechanism that could also allow to predict the effect of doping the material on the barocaloric effects. We respond to his concerns in this letter with already published experimental evidences and we consider that, although very interesting, further theoretical discussions are far from the scope of this concise communication that experimentally presents the **very first example** of an organic-inorganic hybrid compound showing an extraordinary barocaloric effect. And thus opening this large, flexible, multifunctional and designable class of compounds to giant barocaloric behavior.

Reviewer #2 (Remarks to the Author):

The authors have satisfactorily responded to my points of criticism and made major revisions of the manuscript following the suggestions of the referees. The authors have improved the manuscript and satisfactorily replied to most issues raised. Thus, I recommend publication of the article without further changes (there are just some typos along the Supp Information).

We really appreciate that this reviewer thinks that we have satisfactorily responded to all his/her points of criticism and made major revisions of the manuscript following the suggestions of the referees. Thus, he/she recommend publication of the article without further changes.

Reviewer #1 (Remarks to the Author):

The authors courteously answer to my first question about the origin of the entropy change. Now, the readers who are not familiar this type of material can understand the mechanism of this transition.

Meanwhile, in the final paragraph, the explanation is too sloppy. In all the 1st order transition, it is no wonder that entropy change becomes large by increasing degree of order in an ordered state and degree of disorder in a disordered state; this is just a tautology.

As I mentioned in the last comment, if the authors know the mechanism of the transition, then they can estimate (an ideal) maximum entropy change, and they should compare whether the experimental result truly reach that ideal value. Then, if they believe this ideal value can be increased by such as doping

and tailoring, they should explain through what mechanism they can be modified in logical way.

This is also same comment as in the previous report; this material exhibit a largest strength per pressure, but the maximum value at the present stage is not so large (not fully enough for the practical applications).

We thank the comments of the reviewer 1 which states that now “*the readers who are not familiar this type of material can understand the mechanism of this transition*”.

We agree that knowing the mechanism of this complex structural transition could allow estimating a theoretical entropy change that could be compared with the giant experimental value that we have observed by direct methods. We found this recommendation very interesting, but this theoretical estimation is out of the scope of this experimental work that demonstrate a giant barocaloric effect with multiple and consistent experimental techniques. Since the transition mechanism is not fully understood yet, the reviewer 1 is concerned whether the maximum ideal/theoretical value of the entropy change could be increased by doping/tailoring the material. In that regard it should be indicated that, independently of theoretical estimations of the maximum entropy change value derived from a fully understanding of the transition mechanism, it has already been experimentally demonstrated that by changing the Mn (B-cation) of the perovskite by Fe, Co and/or Ni the entropy change associated to the transition decreases [Señarís-Rodríguez, M. A. *et al. J. Mater. Chem. C*, **4**, 4889 (2016)]; Meanwhile when changing the TPrA (A-cation of the perovskite) this entropy increases [Ye, Q. *et al. Journal of Materials Chemistry C*, DOI: 10.1039/C6TC05105G (2017)].

Therefore, it has been experimentally demonstrated that the entropy of this type of systems can be modified by substitution of the A- and/or B- cations.

Moreover, a Nature’s journal (Nature Reviews Materials) has just published a new review (A. K. Cheetham *et al.* (2017), 16099, doi:10.1038/natrevmats.2016.99) where they devoted a complete section of this family of compounds and they state that this type of hybrid organic-inorganic perovskites could be interesting stimuli-induced caloric materials due to their **order– disorder behaviour and framework flexibility** [**“Another interesting avenue would be the study of caloric effects, because the phase transitions in HOIPs [hybrid organic-inorganic perovskites] are often associated with large entropy changes, about an order of magnitude higher than those in oxides^{195,196}. The order– disorder behaviour and framework flexibility could be easily perturbed by external fields, thus leading to large caloric effects.”**].

This recent work remarks the great interest of these materials even if we do not completely understand the complex mechanism of the order-disorder transition.

Of course, we agree with this reviewer in the interest of performing theoretical calculations and deeping into these very complex transition mechanisms. But such a study is out of the scope of the present experimental work that focus on presenting multiple experimental evidences of an unprecedented barocaloric effect in a new family of materials unexplored up-to-date in this context.

On the other hand, the reviewer 1 stated in his previous comments that we “*discuss about very unique results on the barocaloric effect of organic-inorganic compound. The striking feature is that this material exhibit a caloric change comparable to other systems under tremendously small pressure (0.00689 GPa) around room temperature.*” And now, he still agree that “*this material exhibit a largest strength per pressure*” but he thinks that “*the maximum value at the present stage is not so large (not fully enough for the practical applications)*”.

In this context, we have experimentally demonstrated that this [TPrA][Mn(dca)₃] material exhibits a giant barocaloric effect of $\Delta S_{it} \sim 35.1 \text{ J kg}^{-1} \text{ K}^{-1}$ (measured by direct methods and also by quasi-direct methods). This value is much larger than the best magnetocaloric materials and most of the barocaloric materials, many of them good candidates for practical applications, and that have been already used in prototypes [Poredos, A. *et al.*

Magnetocaloric Energy Conversion From Theory to Applications, Springer International Publishing (2015) and ref. therein].

Therefore, we do believe that this unprecedented giant barocaloric results on this organic-inorganic hybrid [TPrA][Mn(dca)₃] has potential interest for practical applications, although further studies and optimizations would of course be still needed.

Reviewer #3 (Remarks to the Author):

Authors have performed the additional calorimetric measurements that were required to calculate reversible barocaloric effects.

- Figure 3, which shows entropy changes up to 38 J K⁻¹ kg⁻¹ that do not correspond to reversible barocaloric effects should therefore not be shown in the manuscript.

Following this recommendation, we have removed Figure 3 from the main manuscript, and have included it as Supplementary Figure 5.

- Following the previous comment, the barocaloric response reported in table 1 should be the response associated with reversible entropy changes, which at 68.9 bar are reduced to 30 J K⁻¹ kg⁻¹.

Following this recommendation, we have included the value of the $\Delta S_{it(rev)}$ in the Table I. Moreover, in order to compare our data with those available in the literature (all the reference data did not take into account the reversibility of the process associated to the thermal hysteresis), we have also included the $\Delta S_{it(direct)}$ and $\Delta S_{it(q-d)}$, using the same criteria of the literature. This is a very common comparison, even in recent publications in Nature Communications [Table 1 of Moya, X. *et al.* Nature Communications 6, 8801 (2015)].

- Authors should display the calorimetric data and all the subsequent analysis in the main paper, both for cooling and heating data, and for all pressures.

Following this recommendations of reviewer 3, we have substituted the former Figure 5 by the former Supplementary Figure 6 (Now Figure 4), which shows calorimetric data and all the subsequent analysis in the main paper, both for cooling and heating data, and for all pressures.

Inspection of the calorimetric data shown previously in figure 2 in the supplementary file reveals that entropy changes while cooling and heating at different pressures are different. These differences will modify the current data analysis, which in some cases seems too simple. Authors should therefore carefully revise their data analysis.

We appreciate the concerns of the reviewer 3 about the results and the data analysis, that although they can seem simple, they are not.

As we had already responded to t reviewer 2 in his previous deep analysis of our work, *“we have rechecked very carefully the value of the entropy change by performing several measurements on several batches of samples, and also by using different instruments and amount of sample. In this context, we have found that the most accurate value of the entropy change of the phase transition is 42.5 J kg⁻¹ K⁻¹. It*

should be noted that this very large entropy change value is very similar to that given in the previous version of the manuscript and to the value reported in **ref. 36** (former **ref. 35** in the previous version of the manuscript: *Inorg. Chem.* **54**, 11680-11687 (2015)), and that the small difference among them can be attributed to experimental errors.”

Moreover, by recommendation of reviewer 2, we had removed the previous Figure 2 because such figure seemed to be quite similar to that of our previous paper (Ref. 36).

As for the concern of reviewer 3 about the difference in the entropy changes on the cooling and heating curves in the low pressure range (shown in the already eliminated Supplementary Figure 2), as we explain in the “Methods” section this is so because the “*temperature control during the cooling ramp is worse than on heating and some deviations were observed.*”

In that context the following Figure A, shows the rate of the different heating and cooling ramps. As it can be observed, the control of the rate of the cooling ramp is worse than that of the heating one (ranging from 2 K min⁻¹ to 1 K min⁻¹). That is the reason why the DSC curves on the previous Supplementary Figure 2 exhibit different shapes in comparison with the heating curves.

Since the isobaric entropy change (ΔS_{ib}) depends on the heating/cooling rate, we have used the real rate of each curve to calculate the ΔS_{ib} at different pressures for both heating and cooling processes. As it can be observed in the Supplementary Figure 4 of the manuscript, once the different rates are considered, all the heating and cooling curves at different pressures exhibit the same value of entropy change ΔS_{ib} .

Figure A. Heating and cooling rate on the low pressure DSC analysis. Temperature (K) vs time (min) of the [TPrA][Mn(dca)₃] sample under the low pressure range.

Also, authors should use the elastic contribution to displace the pressure-dependent entropy curves, instead of the rather arbitrary normalisation that is currently being used.

In the previous version of the manuscript, we had already used the elastic

contribution for the displacement. Nevertheless, we had oversimplified the discussion in the text for the readers who are not familiar this type of analysis, since this journal have a very broad audience working on different fields.

Nevertheless, in this new revised manuscript and following the recommendation of reviewer 3, we have done a more specific discussion about the elastic heating contribution to the displacement of the curves, and we have included a new figure in the supplementary information (Supplementary Figure 7).

This specific discussion can be found in pages. 8 and 9 and in the “Methods” section (Pg. 18) of the revised manuscript, and it has also been indicated in Figures 3, 4 and Supplementary Figure 7.

Supplementary Figure 7. Additional entropy change arising reversibly away from the first-order transition. Temperature dependence of $\Delta S_+(P) = -[m^{-1}(\delta V / \delta T)_{P=0}]P$ on applying pressure P , excluding the temperatures (grey zone) in which the pressure-dependent phase transition occurs.

- The comparison of RCP values divided by pressure, magnetic field and electric field is meaningless, as different applied fields have different units.

We agree with the reviewer 3 that different stimuli units (of pressure, of magnetic of field and electric field) cannot be compared. Nevertheless, the mathematical normalization of the RCP divided by different stimuli units is useful for compare the materials within the same group (i.e. barocaloric materials with themselves). In addition, the comparison within groups could be also useful to have an idea about the amount of magnetic and/or electric field it would be necessary to reach the same RCP in the best magneto- and/or electrocaloric materials related to the best barocaloric materials. Even more, it is very useful for the audience who are not familiar with these stimuli-induced caloric effects. Moreover, tables with this type of caloric effects normalization per stimuli units, such as pressure, are common in the literature [Table 1 of Moya, X. *et al.* *Nature Communications* 6, 8801 (2015)].

- Isothermal measurements: pressure is applied at a much faster rate than it is removed. The sample will be therefore at a higher temperature while applying pressure than while removing pressure, therefore precluding isothermal conditions.

The reviewer 3 is correct in signalling that the difference in the pressure application/removal ramp could lead to non-isothermal conditions.

Nevertheless, it should be recalled that these data have been obtained in a PDSC, equipment which is designed to maintain isothermal conditions by temperature compensation even if the pressure application exhibits a different rate than the pressure release.

Temperature which has been in fact monitored, as shown in the following Figure B.

As it can be seen there is an increase in the temperature associated with the initial increase of the pressure (valves opening) and a temperature decrease in the initial decrease of the pressure (valves closing), but the phase transition does occur at the constant temperature of ~ 331.4 K.

Moreover, the experimental results show exactly the same entropy change either under pressure application and release (even for several cycles), which in addition is consistent with the experimental results observed by other techniques.

Figure B. Direct isothermal entropy change. Calorimetric curves (black solid line) obtained in isothermal conditions (temperature represented by red solid line) under cyclic application/removal of external hydrostatic pressure (blue dash line).

Reviewers' comments:

Reviewer #1 (Remarks to the Author):

The authors exaggerated this referee's comment as if it is demand for construction of an explicit theory. This referee just asked about simple thermodynamic mechanism in standard level. (e.g. , in the water-ice case, the internal energy gain by hydrogen bonding is converted to topological entropy of molecules at the transition, then, their bonding population and the moving degree of freedom governs the limit of the latent heat)

As insisted by the authors in their reply, if one tries to compare just only in viewpoint of the benchmark, by stripping off their material characters or physical ones, then, most of candidate cannot be superior to the pressure induced heat generation at the water-ice transition (and, of course, such an insistence is nonsense.)

In addition, such estimation for the magnitude of caloric phenomenon, even in rough or naïve scale, will support the validity of their experiments.

This referee is not suspicious of their results, but in actual, a certain margin cannot be avoided in estimation of caloric parameters with respect to estimation methods. For example, if one estimates the adiabatic temperature change by using the relation $DT = T(S)_{p1} - T(S)_{p2}$, the value obtained by Fig. 2 is less than or equal to 2 K, which is apparently smaller than the value obtained by the method in the Supplementary section.

To avoid such a quoted or endless arguments, even naïve or rough estimation of magnitude by referring the physical origin can be the last pivotal point for the validity of their data. I hope the authors seriously treat these issues.

Reviewer #3 (Remarks to the Author):

Barocaloric entropy changes calculated using cooling data should be negative, i.e. the entropy decreases on applying pressure.

Barocaloric entropy changes calculated using heating data should be positive.

Authors must correct their results before the paper can accepted for publication.

Author's response to reviewers' comments:

Reviewer #3 (Remarks to the Author):

Barocaloric entropy changes calculated using cooling data should be negative, i.e. the entropy decreases on applying pressure.

Barocaloric entropy changes calculated using heating data should be positive.

Authors must correct their results before the paper can be accepted for publication.

Following the recommendations of reviewer 3, we have modified the barocaloric entropy change signs and accordingly we have also modified the corresponding figures.

Reviewer #2 (Remarks to the Author):

The authors exaggerated this referee's comment as if it is demand for construction of an explicit theory. This referee just asked about simple thermodynamic mechanism in standard level. (e.g., in the water-ice case, the internal energy gain by hydrogen bonding is converted to topological entropy of molecules at the transition, then, their bonding population and the moving degree of freedom governs the limit of the latent heat).

As insisted by the authors in their reply, if one tries to compare just only in viewpoint of the benchmark, by stripping off their material characters or physical ones, then, most of candidate cannot be superior to the pressure induced heat generation at the water-ice transition (and, of course, such an insistence is nonsense).

In addition, such estimation for the magnitude of caloric phenomenon, even in rough or naïve scale, will support the validity of their experiments. This referee is not suspicious of their results, but in actual, a certain margin cannot be avoided in estimation of caloric parameters with respect to estimation methods. For example, if one estimates the adiabatic temperature change by using the relation $\Delta T = T(S)_{p1} - T(S)_{p2}$, the value obtained by Fig. 2 is less than or equal to 2 K, which is apparently smaller than the value obtained by the method in the Supplementary section. To avoid such a quoted or endless arguments, even naïve or rough estimation of magnitude by referring the physical origin can be the last pivotal point for the validity of their data. I hope the authors seriously treat these issues.

We appreciate the comments of this reviewer, which have help us to improve the manuscript.

As for his question about the microscopic physical origin of the change in entropy at this solid-solid transition, in principle it can be made up of configurational, rotational and vibrational contributions. To get physical insight about the factor which is determining the magnitude of the large entropy change observed in [TPrA][Mn(dca)₃], we have made a first estimation of the configurational entropy. For this purpose, we have taken into account the structures of its *polymorphs I* and *II* and the partial order/disorder of the dca and TPrA ion (see Supplementary Figure 2) and used the expression: $\Delta S_{config.} = R \ln(N)$, with $N = (n_2/n_1)$, where n_2 and n_1 are the number of configurations in the two polymorphs (Rao, C.N.R. & Rao, K. J. Phase Transitions in Solids, McGraw Hill (1978), p. 122), obtaining a $N = 10.3$ and a $\Delta S_{conf.} = 19.4 \text{ J mol}^{-1} \text{ K}^{-1}$ ($44.1 \text{ J kg}^{-1} \text{ K}^{-1}$). The good agreement between this entropy value and the experimental one, $\Delta S_{phase} = 42.5 \text{ J kg}^{-1} \text{ K}^{-1}$, (specially taking into account the simplicity of the model and the complexity of the structures) suggests that this configurational factor (the partial order/disorder of the dca and TPrA groups) is mainly determining the total entropy change in this compound.

Supplementary Figure 2. [TPrA][Mn(dca)₃] polymorphs. Structures of the *polymorphs I* and *II* of [TPrA][Mn(dca)₃] obtained by single-crystal XRD and used for calculating the configurational entropy associated to the phase transition using the expression: $\Delta S_{\text{conf.}} = R \ln(N)$, with $N = (n_2/n_1)$, where n_2 and n_1 are the number of configurations in the two polymorphs. Note: the H atoms of the TPrA cations have been removed to facilitate visualization of the structure.

Moreover, the fact that in this hybrid compound the entropy change is mainly determined by the entropy of some atomic arrangements that get partially ordered/disordered, in turn implies that there is plenty of room to get higher values:

(i) by enhancing the configurational entropy: we have estimated that only by obtaining this type of material with a larger degree of order/disorder N can be doubled and $\Delta S_{\text{config.}}$ can be increased by $\sim 28\%$ which would correspond to $\Delta T_{\text{ad}} \sim 7.7$ K, instead of 5 K. Additional possibilities would imply, for example doping a given site of the crystal structure with different types of ions.

(ii) by playing with rotational and vibrational entropies, which at difference with the case of inorganic perovskites can be also relevant in hybrid organic-inorganic perovskites [(i) Butler, K. T., Walsh, A., Cheetham, A. K. & Kieslich, G. Organised chaos: entropy in hybrid inorganic-organic systems and other materials. *Chem. Sci.* **7**, 6316-6324 (2016); (ii) Kieslich, G., Kumagai, S., Butler, K. T., Okamura, T., Hendon, C. H., Sun, S., Yamashita, M., Walsh, A. & Cheetham, A. K. Role of entropic effects in controlling the polymorphism in formate ABX₃ metal-organic frameworks. *Chem. Commun.*, **51**, 15538-15541 (2015)]. For example by including moieties, organic species, with a high tendency to behave as molecular rotators; or by creating materials with stronger interactions between the framework and the species allocated in the cavity (for ex. through H-bonds, etc.).

We have included a shortened version of this discussion in the revised manuscript (Pg. 5 and 15) and we have added a new Supplementary Figure 2.

As for the second part of the question, it should be indicated that the variation of T_c with P shown in Figure 2 corresponds to the so-called temperature span of the material. This is the operational temperature window in which the isothermal entropy change (and the adiabatic temperature change) takes place. [(i) Mañosa, L.I. & Planes, A. Materials with Giant Mechanocaloric Effects: Cooling by Strength. *Adv. Mater.*, 1603607 (2017); (ii) Liu, Y., Wei, J., Janolin, P. E., Infante, I. C., Lou, X. & Dkhil, B. Giant room-temperature barocaloric effect and pressure-mediated electrocaloric effect in BaTiO₃ single crystal. *Appl. Phys. Lett.* **104**, 162904 (2014). (iii) Mañosa, L.I., Jarque-Farnos, S., Vives, E. & Planes, A. Large temperature span and giant refrigerant capacity in elastocaloric Cu-Zn-Al shape memory alloys. *Appl. Phys. Lett.*, **103**, 211904 (2013)].

In fact, such window (thermal span of ~2 K shown in Figure 2) gets broader with pressure, so that for $P = 1000$ bar it reaches a value of ~23 K (see Supplementary Figure 7).

On the other hand, the adiabatic temperature change, ΔT_{ad} , obtained by the method described in page 8 and whose results are shown in Fig. 6b of represents the temperature change of the material itself when an external field (pressure in this case) is applied adiabatically.

A direct determination of the adiabatic temperature change will require to measure the sample temperature by using an appropriate thermometer while the external field is adiabatically modified. Although, this is a quite direct method to quantify a caloric effect, adiabatic temperature change experiments are usually difficult to implement, and the majority of studies are devoted to field-induced isothermal entropy changes, ΔS_{it} [Mañosa, L.I., Planes, A. & Acet, M. Advanced materials for solid-state refrigeration. *J. Mater. Chem. A*, **1**, 4925-4936 (2013)].

When temperature changes ΔT_{ad} are relatively smaller than the temperature at which the stimuli-induced caloric effect occurs, T , ($\Delta T_{ad} \ll T$) and the heat capacity (C_p) can be assumed to be independent of the applied force, a good approximation for obtaining the adiabatic entropy change from isothermal entropy change data is using the relation $\Delta T_{ad} = -T/C_p \times \Delta S_{it}$. [(i) Mañosa, L.I. & Planes, A. Materials with Giant Mechanocaloric Effects: Cooling by Strength. *Adv. Mater.*, 1603607 (2017); (ii) Moya, X., Kar-Narayan, S. & Mathur, N. D. Caloric materials near ferroic phase transitions. *Nat. Mater.* **13**, 439-450 (2014); (iii) Flerov, I. N., Gorev, M. V., Tressaud, A. & Laptash, N. M. Perovskite-like fluorides and oxyfluorides: phase transitions and caloric effects. *Crystallogr. Rep.* **56**, 9-17 (2011)].

REVIEWERS' COMMENTS:

Reviewer #1 (Remarks to the Author):

This referee agrees that the authors closely make answers to all my questions. Now I have no further question and/or objection, and I feel now this manuscript about the unique results becomes an object to broad interests of various readers.

Reviewer #3 (Remarks to the Author):

Editor's note: Reviewer recommended acceptance of the paper and no further comments to the Author were provided.

RESPONSE TO REVIEWERS' COMMENTS:

The authors thank the reviewers for recommending the publication of the manuscript without further changes.

Reviewer #1 (Remarks to the Author):

This referee agrees that the authors closely make answers to all my questions. Now I have no further question and/or objection, and I feel now this manuscript about the unique results becomes an object to broad interests of various readers.

Reviewer #3 (Remarks to the Author):

Editor's note: Reviewer recommended acceptance of the paper and no further comments to the Author were provided.